# Cardiac-specific loss of mitoNEET expression is linked with age-related heart failure

Takaaki Furihata[1], Shingo Takada[1], Naoya Kakutani[1], Satoshi Maekawa[1], Masaya Tsuda[1], Junichi Matsumoto [1], Wataru Mizushima[1], Arata Fukushima[1], Takashi Yokota [1], Nobuyuki Enzan [2], Shouji Matsushima[2], Haruka Handa [3], Yoshizuki Fumoto[3], Junko Nio-Kobayashi[4], Toshihiko Iwanaga[4], Shinya Tanaka[5], Hiroyuki Tsutsui[2], Hisataka Sabe[3] & Shintaro Kinugawa [1✉]

Heart failure (HF) occurs frequently among older individuals, and dysfunction of cardiac mitochondria is often observed. We here show the cardiac-specific downregulation of a certain mitochondrial component during the chronological aging of mice, which is detrimental to the heart. MitoNEET is a mitochondrial outer membrane protein, encoded by CDGSH iron sulfur domain 1 (CISD1). Expression of mitoNEET was specifically downregulated in the heart and kidney of chronologically aged mice. Mice with a constitutive cardiac-specific deletion of CISD1 on the C57BL/6J background showed cardiac dysfunction only after 12 months of age and developed HF after 16 months; whereas irregular morphology and higher levels of reactive oxygen species in their cardiac mitochondria were observed at earlier time points. Our results suggest a possible mechanism by which cardiac mitochondria may gradually lose their integrity during natural aging, and shed light on an uncharted molecular basis closely related to age-associated HF.

[1] Department of Cardiovascular Medicine, Faculty of Medicine and Graduate School of Medicine, Hokkaido University, Sapporo, Japan. [2] Department of Cardiovascular Medicine, Kyushu University Graduate School of Medical Sciences, Fukuoka, Japan. [3] Department of Molecular Biology, Faculty of Medicine and Graduate School of Medicine, Hokkaido University, Sapporo, Japan. [4] Laboratory of Histology and Cytology, Department of Anatomy, Faculty of Medicine and Graduate School of Medicine, Hokkaido University, Sapporo, Japan. [5] Department of Cancer Pathology, Faculty of Medicine and Graduate School of Medicine, Hokkaido University, Sapporo, Japan. ✉email: tuckahoe@med.hokudai.ac.jp

Heart failure (HF) is a common disease among older people, and is a major public health issue worldwide. Age-associated HF is frequently characterized by cardiac hypertrophy and interstitial fibrosis, in which oxidative stress and inflammation, as well as protein misfolding and cell death, are well recognized as major causes[1]. On the other hand, the heart is rich in mitochondria[2], and mitochondrial dysfunction is another mechanism closely associated with HF[3–5]. Mitochondria are the major source of intracellular reactive oxygen species (ROS) via the leak of electrons from oxidative phosphorylation systems. Thus, although ROS demonstrate duality in cellular functions, the accumulation of ROS-associated damage in cardiac mitochondria during aging is likely to cause further dysfunction of the heart[6,7]. However, the complete mechanism by which the integrity of cardiac mitochondria become affected during aging still remains elusive.

MitoNEET, encoded by CDGSH iron sulfur domain 1 (CISD1), is a protein localized to the outer membrane of mitochondria, which was originally identified as a target of the insulin-sensitizing drug pioglitazone[8]. MitoNEET contains redox-sensitive 2Fe-2S clusters[9,10], and under oxidized state, it can transfer its 2Fe-2S clusters to an apo-acceptor protein[11–13]. Recently, it has been shown that mitoNEET plays a critical role in cytosolic Fe-S cluster repair of iron-regulatory protein-1[14]. MitoNEET can also regulates free iron level within mitochondria by regulating the channel function of voltage-dependent anion channel 1[15]. The systemic inducible knockdown of mitoNEET was found to cause mitochondrial iron overload in adipocytes and liver, and increased iron-mediated mitochondrial lipid peroxidation[16]. MitoNEET may also function as an electron-transfer protein[9]. However, the exact roles of this protein presently remain elusive.

To investigate the uncharted mechanisms by which cardiac mitochondria may lose their integrity during aging, we first analyzed the levels of mitochondrial proteins and found that the mitoNEET protein is specifically downregulated in the hearts of normally aged mice. Other mitochondrial outer membrane proteins, MITOL, Tom22, Tom40, and Tom70, did not show such a decrease. A similar age-dependent downregulation of mitoNEET was observed in the kidney, but not in the liver, skeletal muscle, or brain. By generating genetically engineered mice in which mitoNEET (i.e., CISD1) is constitutively deleted in a manner specific to cardiac muscle, we found that the loss of mitoNEET only gradually, but not immediately, affects the functions of the heart. Our results hence demonstrate a possible natural mechanism by which cardiac mitochondria gradually lose their integrity during chronological aging, which is related to age-associated HF.

## Results

### mitoNEET expression is specifically downregulated in the aged mouse heart.
We first analyzed the levels of mitochondrial proteins during chronological aging. The amount of mitochondria per weight of cardiac muscle may change during aging[17]. We found that mitoNEET levels are significantly decreased in the hearts of 12-month-old C57BL/6J mice compared with those of 3-month-old mice, in which relative protein levels were normalized to total protein evaluated by Coomassie Brilliant Blue (CBB) staining (Fig. 1). MITOL, Tom22, Tom40, and Tom70, which are outer mitochondrial membrane protein like mitoNEET, did not show such a decrease (Fig. 1). Miner1, which is a mitoNEET homolog and is localized to the mitochondria-associated membrane, also did not change during aging (Fig. 1). A similar age-dependent decrease in mitoNEET levels was also observed in the kidney, but not in the liver, the skeletal muscle, or the brain

(Supplementary Fig. 1). Therefore, these results indicated that certain mitochondrial components become selectively downregulated in chronologically aged mice, particularly in the heart and the kidney.

### Cardiac-specific mitoNEET knockout mice demonstrate left ventricular dysfunction and death from HF after chronological aging.
To analyze the possible effects of the decrease in mitoNEET levels on cardiac function, we then generated cardiac-specific mitoNEET (i.e., CISD1) gene knockout (KO) mice using the general lox-P and homologous recombination strategy. The mitoNEET targeting construct and the genomic structure of mitoNEET are shown in Supplementary Fig. 2a. MitoNEET flox/flox mice were used as a control and were crossed with αMHC-Cre mice to obtain mice with cardiac-specific deletion of mitoNEET (Supplementary Fig. 2b). We first confirmed that CISD1 mRNA was undetectable in the hearts of these KO mice (Supplementary Fig. 2c). Immunoblot and immunohistochemical analysis also confirmed absence of the mitoNEET protein in cardiac cells of KO mice (Supplementary Fig. 2d, e), whereas it was expressed in the brain, heart, liver, kidney, and skeletal muscle of KO mice, as in control mice (Supplementary Fig. 2f). There was no significant difference in the expression of Miner1 in the heart between KO and control (Supplementary Fig. 2g). These cardiac-specific mitoNEET KO mice were viable and fertile, and there were no notable differences in appearance and body weight between KO mice and control mice (Supplementary Table 1).

Left ventricular (LV) function and morphology of the mice were then evaluated by echocardiography. No significant differences were detected with regard to all echocardiographic parameters between KO mice and controls at 3 months of age (Fig. 2a–e). However, the LV end-diastolic diameter and LV end-systolic diameter were substantially higher in KO mice than in controls, when measured at 12 months of age (Fig. 2a–c). In these older mice, the fractional shortening was also significantly lower in the KO mice than in the controls (Fig. 2d), whereas the LV wall thickness was almost comparable between the two groups, even when they were older than 12 months (Fig. 2e). Moreover, values of LV weight per body weight also became significantly higher in KO mice than in the controls, when they were older than 12 months (Supplementary Table 1). Cardiac interstitial fibrosis is a characteristic of the hearts of aged organisms[1]. Larger areas of interstitial fibrosis of the heart were observed in KO mice than in controls at 12 months of age, but not at 3 months of age (Fig. 2f, g). On the other hand, lung weight per body weight was significantly higher in KO mice than in controls, only after they were older than 16 months (Supplementary Table 1). Therefore, the detrimental effects of constitutive cardiac mitoNEET deletion from birth appeared to become evident as LV dysfunction when mice became aged, such as older than 12 months; and this dysfunction appeared to be followed by the development of HF after another several months. Kaplan–Meier analysis then demonstrated that a substantial number of mitoNEET KO mice start to die after 15 months of age, and none of them survived for longer than 22 months, whereas all of the control mice lived for longer than 22 months (Fig. 2h). Therefore, these results collectively suggested that cardiac mitoNEET deletion gradually, rather than immediately, affects the heart as well as overall life activity.

### Cardiac mitochondrial respiratory activities are impaired in mitoNEET KO mice after 12 months of age.
We next analyzed mitochondrial function of cardiac muscle cells isolated from mitoNEET KO mice and control mice. Mitochondria produce adenosine triphosphate (ATP) by the oxidative phosphorylation

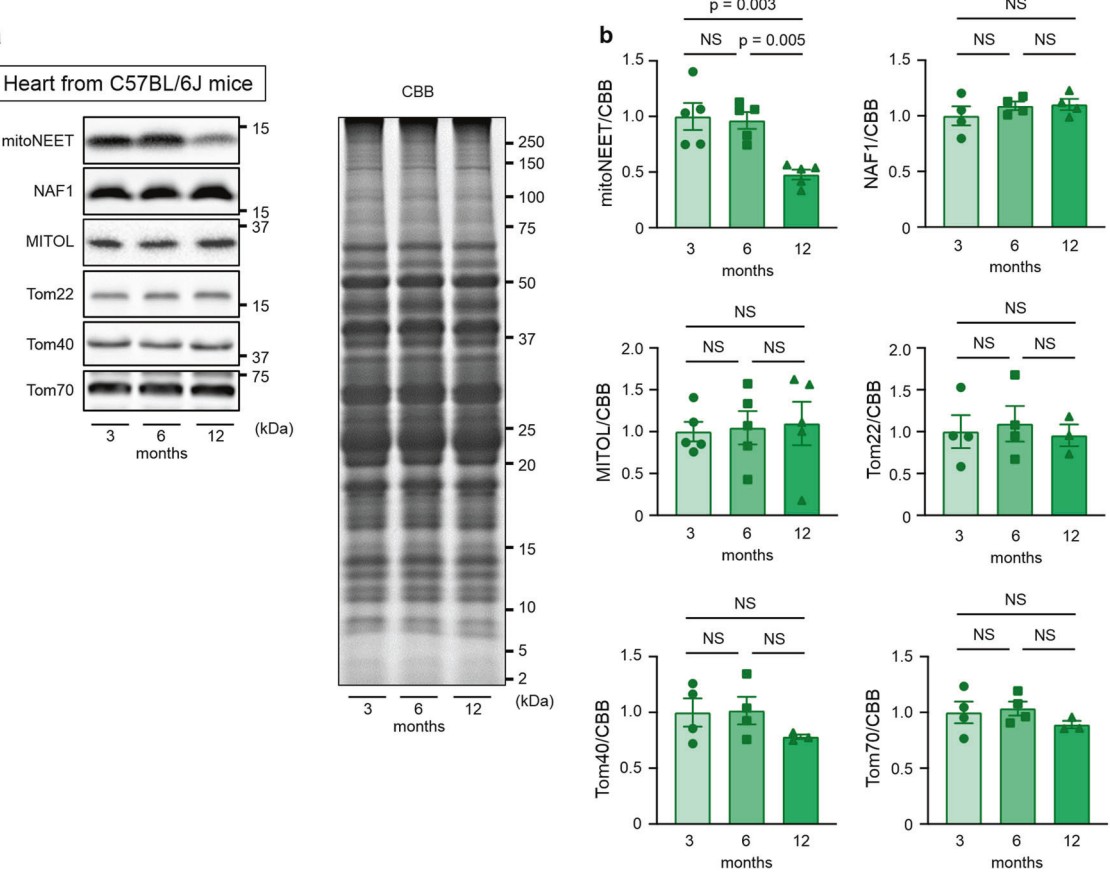

**Fig. 1 MitoNEET expression is specifically downregulated in the hearts of aged mice.** Representative immunoblot (**a**) and summary data (**b**) of mitoNEET, Miner1, MITOL, Tom22, Tom40, and Tom70 protein expression normalized to total protein evaluated by CBB staining in the hearts of C57BL/6J mice at 3 months ($n = 5$ for mitoNEET and MITOL, $n = 4$ or 3 for others), 6 months ($n = 5$ for mitoNEET and MITOL, $n = 4$ or 3 for others), and 12 months ($n = 5$ for mitoNEET and MITOL, $n = 4$ or 3 for others) of age. Data are shown as the mean ± s.e.; individual data points are shown. ANOVA was performed followed by the Tukey test for comparisons of means among 3 groups. CBB, Coomassie Brilliant Blue; NS, not significant.

system, which consists of complexes I–IV of the electron transport chain. Mitochondrial respiration is classified into five different states by the specific rate-limiting component, and the respiratory chain is rate-limiting in state 3. It was shown previously by the analysis of systemic mitoNEET-null mice that complex I-dependent state 3 respiration is significantly decreased in cardiac mitochondria in the absence of mitoNEET[8]. However, we found that state 3 respiration, as well as maximal respiratory capacity and reserve respiratory capacity, are almost comparable between KO mice and controls, when measured at 3 months of age (Fig. 3). On the other hand, all of these respiratory activities were significantly impaired in KO mice compared with controls, when they were older than 12 months (Fig. 3). Reserve respiratory capacity is crucial for promoting cell survival[18]. Of note, the reserve respiratory capacity became substantially low in the cardiac mitochondria of KO mice at 16 months of age (Fig. 3c). Therefore, the loss of mitoNEET from birth caused detrimental effects on mitochondrial respiration only when mice became old, as we observed in the 12-month-old KO mice. On the other hand, there were no significant differences in state 2 respiration between controls and KO mice at all ages (Fig. 3). State 4 respiration was slightly decreased in KO mice compared with controls, when they were older than 12 months (Fig. 3).

**_mitoNEET_ KO affects morphological integrity of mitochondria.** Transmission electron microscopic analysis demonstrated

no notable differences between the _mitoNEET_ KO mice and controls with regard to the number of mitochondria and their cross-sectional areas, when analyzed at 3, 12, and 16 months of age (Fig. 4). In contrast, cristae density was significantly decreased in KO mice compared to controls, when measured at 3 and 12 months of age (Fig. 4). Morphological integrity, as well as integrity of the outer membrane of mitochondria, is crucial to avoid the unnecessary leakage of electrons from the respiratory chain, that otherwise leads to the generation of ROS[3,19]. $H_2O_2$ release rates from mitochondria were already higher in KO mice than in controls at 3 months of age (Fig. 5a, b). Together with the above results, these results collectively suggested that the loss of mitoNEET may primarily affects the morphological integrity of mitochondria, and this becomes evident at much earlier ages in mice than the dysfunction of mitochondrial respiration and LV function. We also evaluated 4-hydroxy-2-nonenal (HNE), an aldehydic byproduct of lipid peroxidation, in the heart of controls and mitoNEET KO mice during aging. HNE was increased in the heart from mitoNEET KO mice compared with controls both at 3 months and 12 months of age (Fig. 5c, d). Changes in HNE was consistent with the production of $H_2O_2$. Long-term accumulation of oxidative damage due to excessive production of ROS would be associated with the dysfunction of mitochondrial respiration and LV function.

MitoNEET contains an 2Fe-2S cluster[9,10]. The iron content of mitochondria isolated from cardiac muscle was substantially higher in _mitoNEET_ KO mice than in control mice, both at

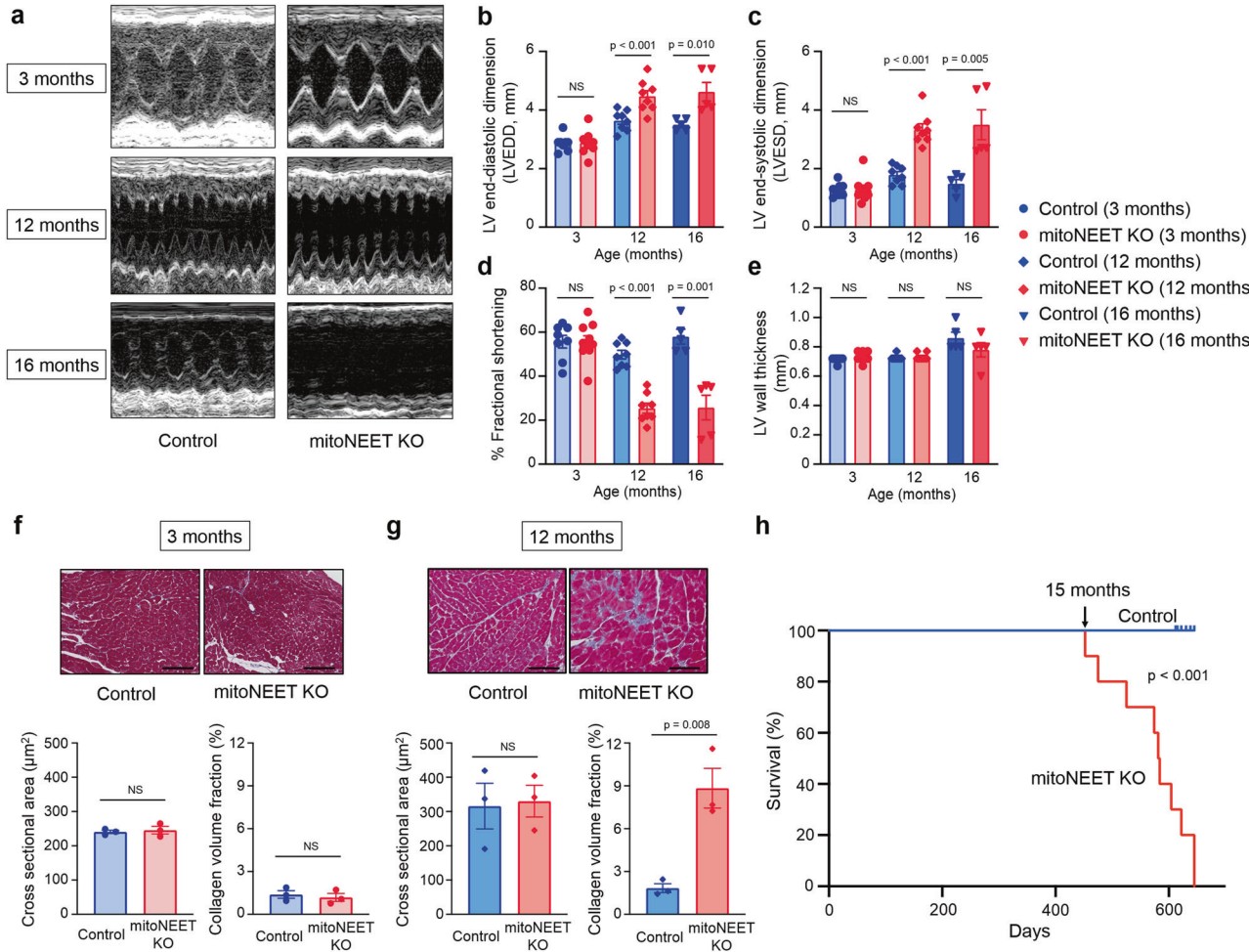

**Fig. 2 Cardiac-specific *mitoNEET* KO mice demonstrate LV dysfunction, HF, and HF death with chronological aging.** Representative echocardiographic images (**a**) and summary data of left ventricular end-diastolic dimension (LVEDD) (**b**), left ventricular end-systolic dimension (LVESD) (**c**), % fractional shortening (**d**), and LV wall thickness (**e**) in Control and mitoNEET KO mice at 3 months ($n = 8$ and 9, respectively), 12 months ($n = 8$ and 9, respectively), and 16 months of age ($n = 5$ each). Representative Masson trichrome staining and summary data of the cross-sectional area and collagen volume fraction of Control and mitoNEET KO mice at 3 months (**f**; $n = 3$ each) and 12 months (**g**; $n = 3$ each) of age. Scale bar, 100 μm. Kaplan–Meier survival curves of Control ($n = 10$) and mitoNEET KO mice ($n = 10$) (**h**). Arrow indicates the point of 15 months. Data are shown as the mean ± s.e.; individual points are shown. The Student unpaired two-tailed *t*-test was performed to compare means between Control and mitoNEET KO mice of identical age. The differences between Control and mitoNEET KO mice in survival were tested by the log-rank test. NS, not significant.

3 months and 12 months of age (Fig. 5e, f). The amount of mitochondrial ferritin (FtMt), which is an iron-storage protein, was also substantially higher in the KO mice than controls (Supplementary Fig. 3a). In contrast, in isolated mitochondria, the levels of proteins that are involved in mitochondrial iron homeostasis, such as mitoferrin-2 (MFRN2), frataxin (FXN), ATP-binding cassette protein B7 (ABCB7), and ATP-binding cassette protein B8 (ABCB8) were not significantly different between the KO mice and controls (Supplementary Fig. 3b–e). The levels of proteins involved in cytosolic iron homeostasis, such as transferrin receptor (TfR), divalent metal transporter 1 (DMT1), and ferroportin (Fpn) did not differ between KO mice and controls (Supplementary Fig. 3f–h). This was also the case with proteins primarily involved in cellular iron homeostasis, such as iron regulatory protein 1 (IRP1) and iron regulatory protein 2 (IRP2) (Supplementary Fig. 3i, j). Moreover, the amounts of heme and proteins involved in heme synthesis were not increased in whole hearts and in the myocardial mitochondria of KO mice compared with controls (Supplementary Fig. 3k–n). Therefore, the loss of mitoNEET appears to increase

the amounts of iron and its storage protein in mitochondria, without notably affecting the levels of proteins that are involved in the transport and metabolism of iron in mitochondria and the cytosol, although the precise molecular mechanisms therein involved still remain unclear.

## Discussion
In this study, although the amounts of mitochondria in cardiac muscles are known to change during aging[17], we showed that mitoNEET protein expression is specifically downregulated in the hearts of chronologically aged mice. However, the precise mechanism involved in this age-associated mitoNEET downregulation still remains unclear, although we observed that *mitoNEET* mRNA levels were slightly decreased in the hearts of aged mice (our unpublished results). Recent paper reported significant increases in Parkin, an E3 ubiquitin ligase, and ubiquitinated proteins in the mitochondrial fraction from hearts of C57BL/6 mice at 24 months of age[20]. mitoNEET has been shown to be a direct substrate of Parkin and to be polyubiquitinated by Parkin[21]. Therefore, Parkin may be involved in degradation of

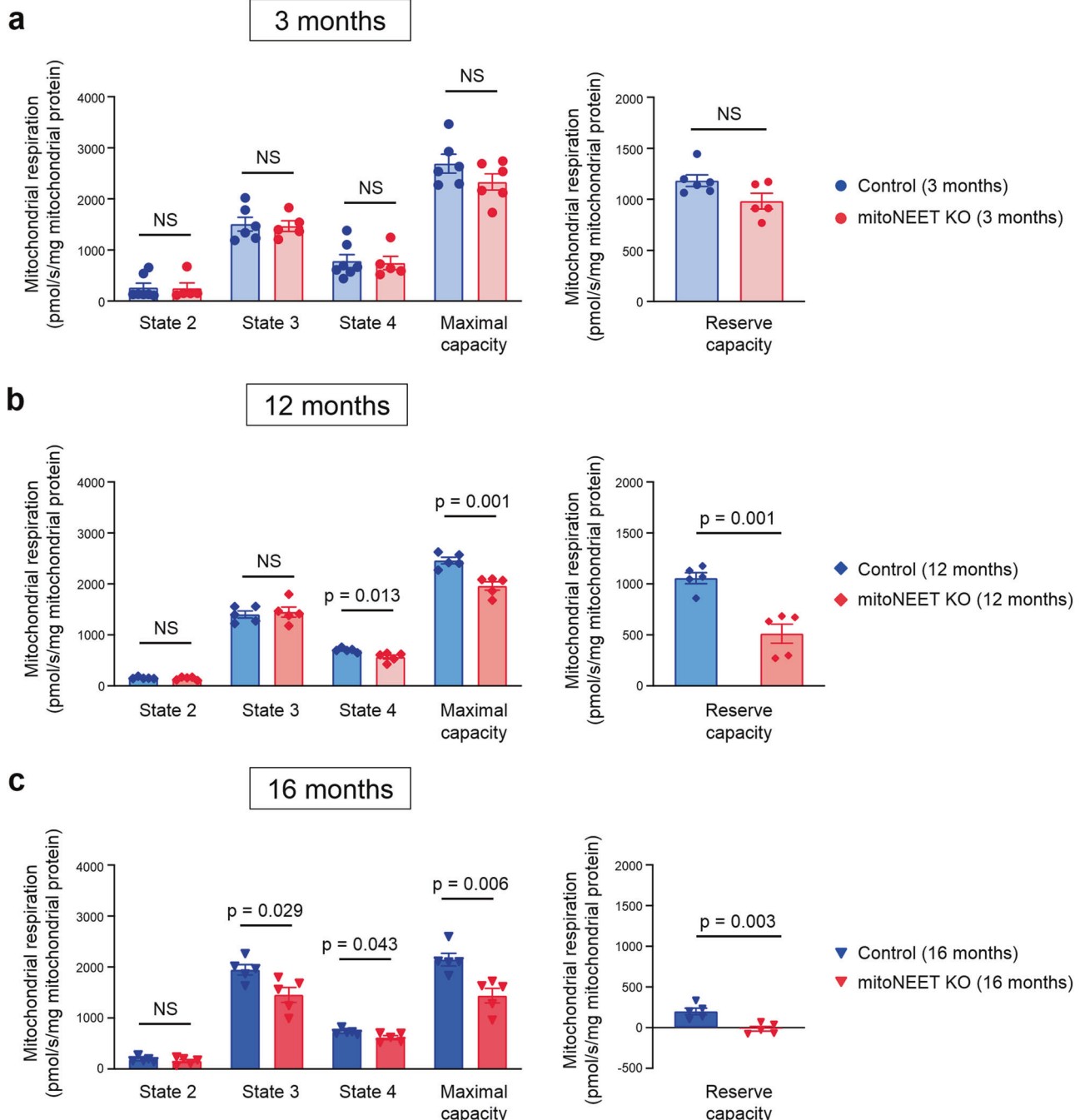

**Fig. 3 Cardiac mitochondrial respiratory activities are impaired in *mitoNEET* KO mice older than 12 months.** Mitochondrial respiration in isolated mitochondria during state 2, state 3, state 4, maximal capacity, and reserve capacity of control and mitoNEET KO mice at 3 months (**a**; $n = 6$ each), 12 months (**b**; $n = 5$ each), and 16 months of age (**c**; $n = 5$ each). Data are shown as the mean ± s.e.; individual data points are shown. The Student unpaired two-tailed *t*-test was performed to compare means between Control and mitoNEET KO mice of identical age. NS, not significant.

mitoNEET in aged hearts, however, the functional implications of this translational modification of mitoNEET are unknown.

On the other hand, it is noteworthy that a similar downregulation of mitoNEET is seen in the kidney but not in other tissues and organs. The heart and kidney are both rich in mitochondria, as well as in vascular systems, and are hence highly susceptible to the activation of systemic and local neurohormonal factors and oxidative stress[22–24]. These two organs are closely associated with each other, particularly with regard to their essential roles in regulating homeostasis of the whole body, and

are hence crucial in determining the lifespan of individuals. Whether the age-dependent downregulation of mitoNEET in the heart and the kidney is an intrinsically programmed biological mechanism, or merely the result of some adverse effects specific to these organs, such as hyperoxidative stress, deserves further experimental scrutiny. Furthermore, whether the same mechanism as mice operates in humans also awaits to be determined, together with clarification of the exact roles of mitoNEET in mitochondrial functions. Whether mitochondrial proteins other than mitoNEET also undergo similar age-dependent

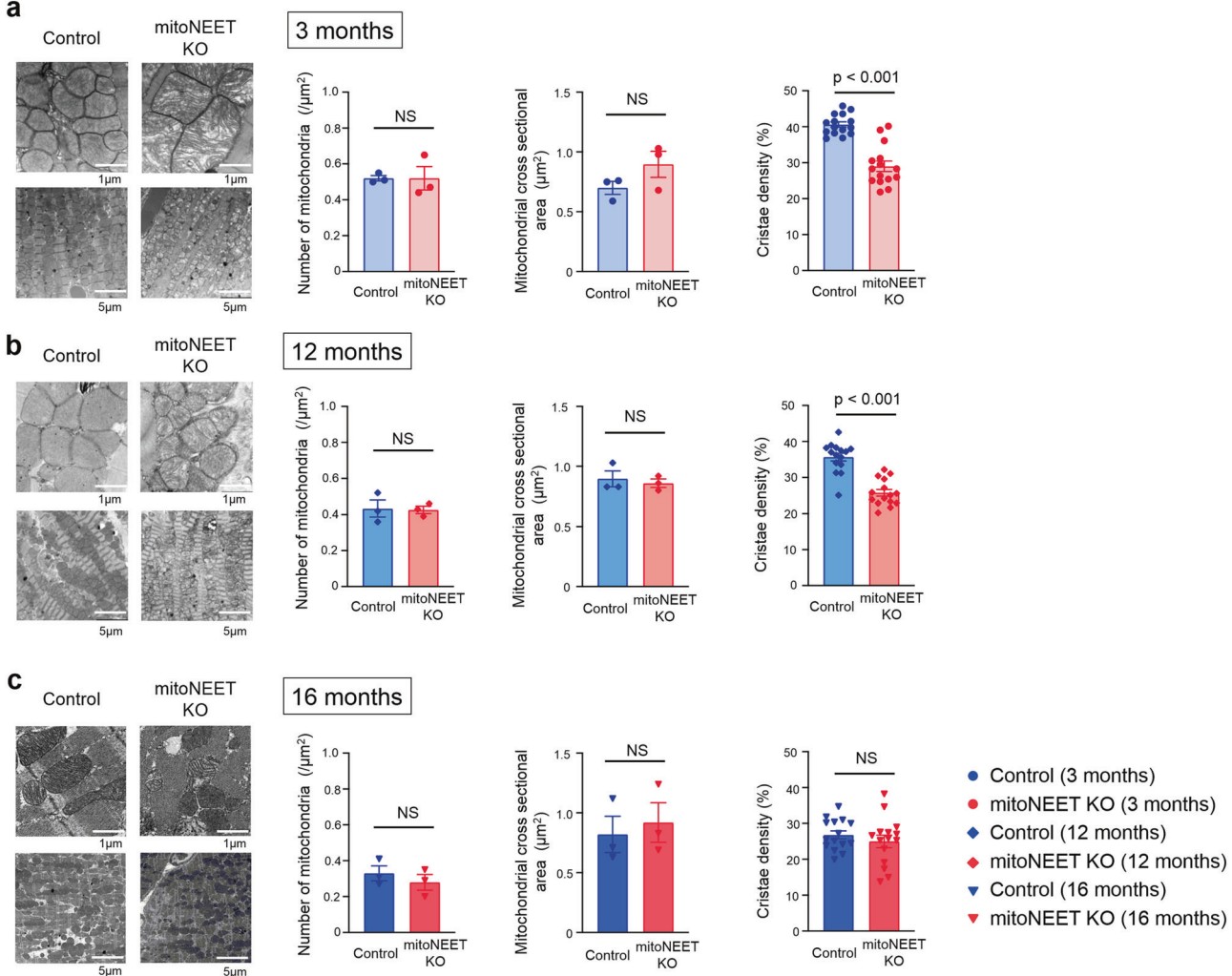

**Fig. 4 MitoNEET KO mice have abnormal mitochondrial morphological integrity.** Representative electron microscopic images and summary data of the number of mitochondria, mitochondrial size, and cistae density in Control and mitoNEET KO mice at 3 months (**a**; n = 3 each), 12 months (**b**; n = 3 each), and 16 months of age (**c**; n = 3 each). Data are shown as the mean ± s.e.; individual data points are shown. The Student unpaired two-tailed t-test was performed to compare means between Control and mitoNEET KO mice of identical age. NS, not significant.

downregulation, which gradually, rather than immediately, disrupts the function of the heart should also be clarified.

Under the condition of mitochondrial iron overload with the deletion of mitoNEET, an increase in $H_2O_2$ release was detected. Iron plays a crucial role in the redox reaction in vivo, and its overload can cause free radical production through many pathways via a reduction of oxygen. Moreover, ROS are generated by means of the Fenton reaction in the presence of endogenous iron. Therefore, a mitochondrial iron overload can easily enhance superoxide production via oxidative phosphorylation, even if the overall mitochondrial function is preserved. This was consistent with the overall mitochondrial function and cardiac function in the 3-month-old mice. Furthermore, it has been known that mitoNEET functions as electron-transfer protein[9,25–27]. The reduced flavin mononucleotide interacts with mitoNEET via specific binding site and transfers its electrons to the 2Fe-2S clusters of mitoNEET, and the reduced 2Fe-2S clusters in mitoNEET transfer the electrons to oxygen or ubiquinone. Therefore, mitoNEET may participate in ROS metabolism in mitochondria. These suggest that disruption of mitoNEET primarily causes a mitochondrial iron overload and enhances ROS production, which secondarily leads to mitochondrial dysfunction (Fig. 6). Enhanced ROS production leads to cardiac dysfunction and the

development of HF, resulting in short life span (Fig. 6). We reported that the exposure of cardiac myocytes to $H_2O_2$ to leads to their injury[28]. We also demonstrated that mitochondria-derived ROS production was increased in hearts from HF model mice[29], and that treatment with an antioxidant and the over-expression of a mitochondrial antioxidant (such as peroxir-edoxin-3) improved cardiac function and HF[30]. Therefore, a long-term exposure of ROS over the physiological level could lead to cardiac dysfunction.

mitoNEET was originally identified as a target of the insulin-sensitizing drug pioglitazone. It has been known that binding of pioglitazone stabilizes mitoNEET against 2Fe-2S cluster release, which are observed under oxidized condition[10,12]. Pioglitazone can inhibit iron transfer from mitoNEET to mitochondria, and iron overload within mitochondria[12]. Furthermore, pioglitazone treatment for spinal cord injury mice improved state 3 respiration and electron transport system capacity of mitochondria in neuron cells, and these effects of pioglitazone were not observed in mitoNEET KO mice[31]. Therefore, our data suggest that long-term treatment with pioglitazone may prevent age-associated cardiac dysfunction and heart failure by stabilizing mitoNEET, and age-associated mitoNEET downregulation may weaken its effectiveness.

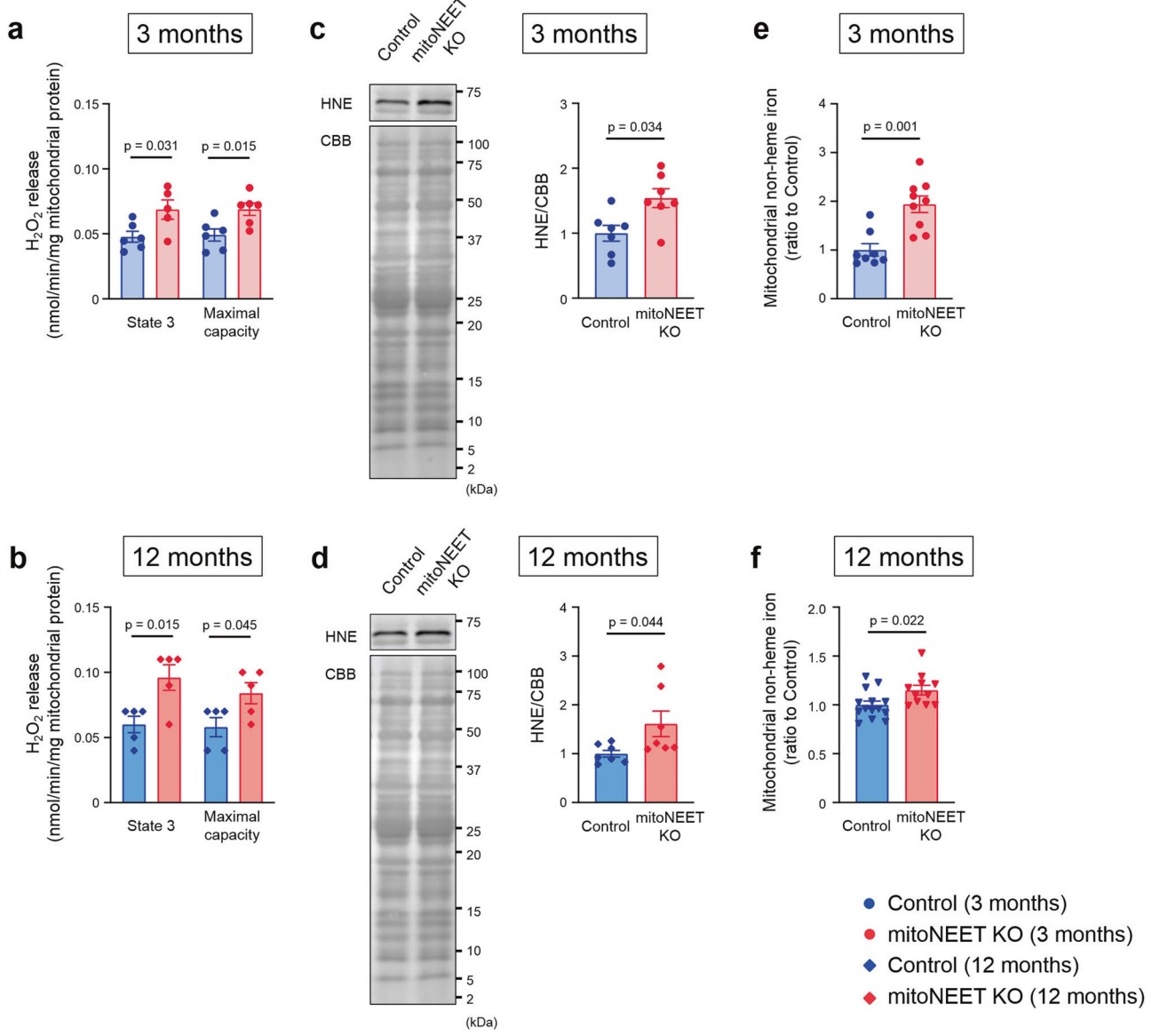

**Fig. 5 *MitoNEET* KO mice have abnormal mitochondrial quality control.** $H_2O_2$ release during mitochondrial state 3 respiration and maximal respiratory capacity in control mice and mitoNEET KO mice at 3 months (**a**; $n = 6$ each) and 12 months of age (**b**; $n = 5$ each). Representative immunoblot and summary data of HNE protein levels in the heart from control mice and mitoNEET KO mice at 3 months (**c**; $n = 7$ each) and 12 months of age (**d**; $n = 7$ each). Mitochondrial nonheme iron content in control mice and mitoNEET KO mice at 3 months (**e**; $n = 8$ and 9, respectively) and 12 months of age (**f**; $n = 14$ and 11, respectively). Data are shown as the mean ± s.e.; individual data points are shown. The Student unpaired two-tailed *t*-test was performed to compare means between Control and mitoNEET KO mice of identical age. CBB, Coomassie Brilliant Blue; NS, not significant; HNE, 4-hydroxy-2-nonenal.

In summary, our study demonstrated an uncharted mechanism by which the downregulation of mitoNEET, which occurs in normal, chronologically aged mice, causes the gradual impairment of cardiac functions during aging (Fig. 6). It is clear that each species of vertebrates, including *Homo sapiens*, has its own average lifespan that is thought to be closely associated with functions of the heart, particularly with its beating[32]. Solving the above issues may contribute to the identification of the molecular machinery that determines the average lifespan of higher animals, including humans, together with possible molecular mechanisms to slow down the aging process.

## Methods

All experimental procedures and methods of animal care were approved by the Institutional Animal Care and Use Committee of National University Corporation Hokkaido University (Permit no. 16-0101) and conformed to the Guide for the

Care and Use of Laboratory Animals published by the U.S. National Institutes of Health.

**Experimental protocols**. All mice were bred in a pathogen-free environment and housed in an animal room under controlled conditions on a 12 h:12 h light/dark cycle at a temperature of 23–25 °C. The experiments were not randomized. The investigators were blinded to the allocation of the mice during certain experiments and outcome assessments.

**Experiment 1: age-dependent reduction of mitoNEET in the heart**. Male C57BL/6J mice (CLEA Japan, Tokyo) were euthanized under deep anesthesia with an overdose of pentobarbital (100 mg/kg i.p.) at the age of 3, 6, and 12 months. The heart, kidney, liver, gastrocnemius muscle, and whole brain were excised, and used for immunoblot analysis.

**Experiment 2: role of mitoNEET in age-dependent cardiac dysfunction**. Mice with a disruption of cardiac-specific *mitoNEET* were generated (Supplementary Fig. 2a). For generation of the conditional allele, we used the specific *mitoNEET*

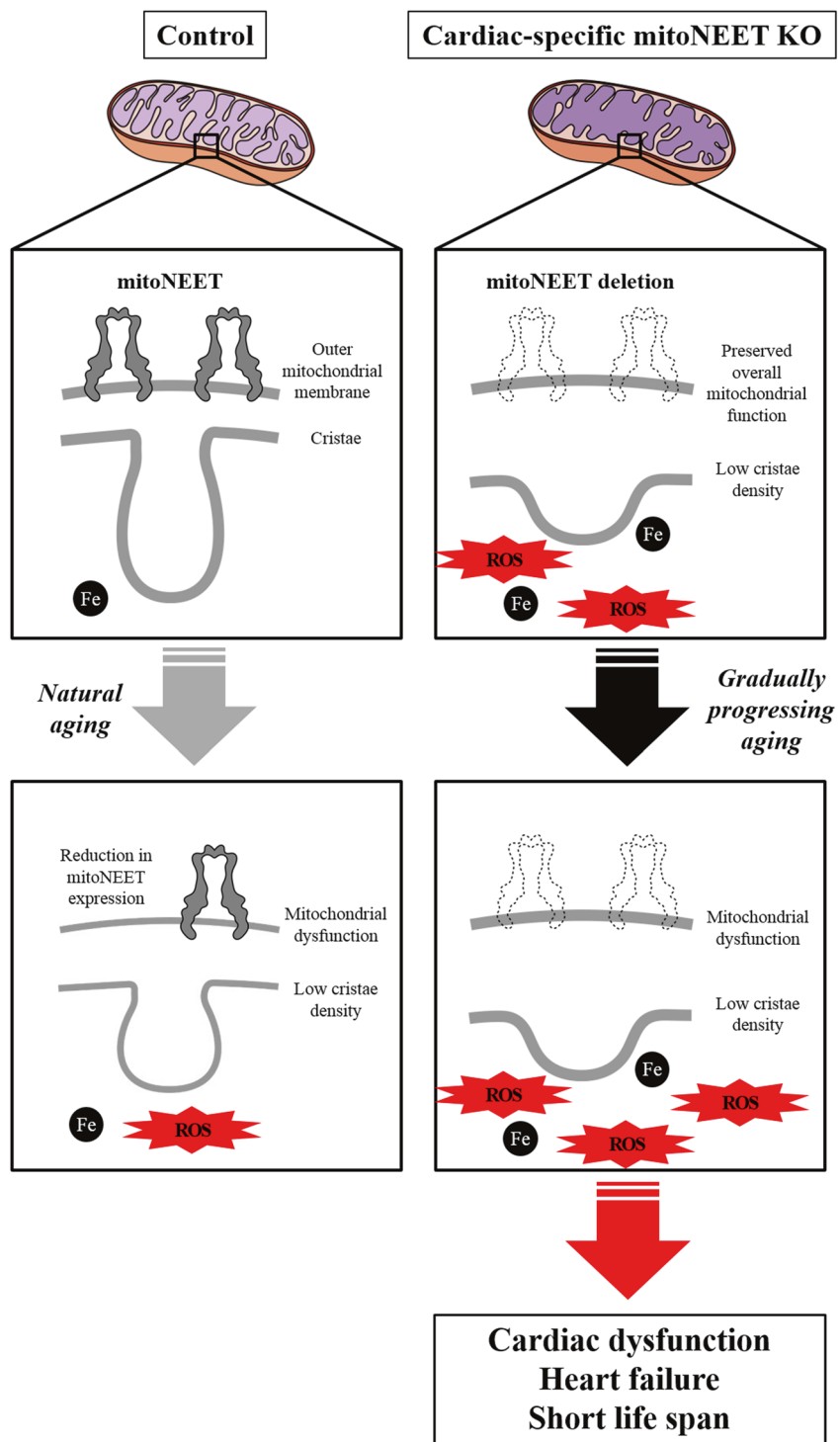

**Fig. 6 Proposed mechanisms for causing age-associated heart failure by cardiac-specific mitoNEET downregulation.** The mitoNEET protein is specifically downregulated in the hearts during normal aging. Deletion of mitoNEET primarily causes an impairment of mitochondrial integrity and a mitochondrial iron overload, and enhances ROS production, which secondarily leads to mitochondrial dysfunction. Enhanced ROS production leads to cardiac dysfunction and the development of heart failure, resulting in short life span. ROS, reactive oxygen species.

open reading frame (ORF): ATG GGC CTC AGC TCC AAC TCC GCT GTG CGA GTT GAG TGG ATC GCG GCC GTC ACC TTT GCT GCT GGC ACA GCC GCT CTC GGT TAC CTG GCT TAC AAG AAG TTC TAC GCT AAA GAG AAT CGC ACC AAA GCT ATG GTG AAT CTT CAG ATC CAG AAA GAC AAC CCG AAG GTG GTG CAT GCC TTC GAC ATG GAG GAT CTG GGG GAT AAG GCC GTG TAC TGC CGA TGC TGG AGG TCT AAA AAG TTC CCC TTC TGC GAT GGG GCT CAC ATA AAG CAC AAC GAA GAG ACT GGC GAC AAC GTA GGA CCT CTG ATC ATC AAG AAA AAG GAA ACC

TAA. Briefly, a conditional *mitoNEET* allele was created using clones isolated by restriction mapping of a genome library, and a C57BL/6 J genetic background clone was used to construct the targeting vector. A neo cassette was inserted downstream of exon 2 and was flanked by FRT sites for later excision by FLP recombinase. Lox-P sites were inserted upstream of exon 2 and downstream of the neo cassette. The targeting vector was transfected by electroporation of embryonic stem cells. After selection, the surviving clones were expanded for PCR analysis to identify recombinant embryonic stem clones. Targeted embryonic stem cells were

microinjected into C57BL/6J blastocysts, and chimeric mice were mated with wild-type C57BL/6J homozygous FLP mice to remove the neo cassette. Heterozygous mice with a neo deletion and confirmed lox-P sites were crossed with C57BL/6J mice (Supplementary Fig. 2b) to obtain heterozygous mice. Finally, *mitoNEET* flox/flox mice were also crossed with αMHC-Cre mice (The Jackson Laboratory, Bar Harbor, ME) to obtain cardiac-specific *mitoNEET* KO mice. *mitoNEET* flox/flox mice were used as a control. Experiments were performed using both male and female Control and *mitoNEET* KO mice of 3, 12, and 16 months of age.

**Experiment 3: survival analysis of *mitoNEET* KO mice.** Survival analysis was performed on another set of male Control and *mitoNEET* KO mice. Starting at 4 weeks of age, cages were inspected daily for dead animals until all of the *mitoNEET* KO mice had died.

**Genotyping of *mitoNEET* KO mice.** Genotyping of *mitoNEET* KO mice was performed by PCR with DNA extracted from the tail. To detect Cre recombinase, the following primers were used: 5′-CTGAAAAGTTAACCAGGTGAGAATG-3′ (forward) and 5′-AGGTAGTTATTCGGATCATCAGCTA-3′ (reverse). To distinguish between mitoNEET flox/flox and wild-type mice, the following primers were used: 5′-TCTAAAATGTACAGCAGCCATGAAG-3′ (forward) and 5′-ACCAAGATACTTAGCGGTAGAAGTG-3′ (reverse). The protocol foe PCR amplification was as follows: 35 cycles of 10 s at 98 °C, 5 s at 65 °C, and 120 s at 72 °C; followed by 35 cycles of 10 s at 98 °C, 5 s at 66 °C, and 60 s at 72 °C.

**Generation of the mitoNEET antibody.** The peptide antigen, which was a C-terminal fragment of mitoNEET, was chemically synthesized. The peptide was injected into 4-month-old female New Zealand white rabbits. After multiple immunizations, blood samples were collected from the ear vein of the rabbits. The specificity of the resulting antibody for mitoNEET was confirmed by immunoblotting following Tris-Tricine sodium dodecyl sulfate-polyacrylamide gel electrophoresis (SDS-PAGE) of control mouse heart lysates. The C-terminal fragment of mitoNEET (just below 2 kDa) was used as a positive control (Supplementary Figure 2c).

**Echocardiography.** Echocardiographic measurements were performed in the conscious state to avoid any effects of the anesthesia on cardiac function. A commercially available echocardiography system (Aplio™300, Toshiba Medical Systems, Odawara, Japan) was used with a dynamically focused 12-MHz linear array transducer and a depth setting of 2.0 cm. A two-dimensional parasternal short-axis view was obtained at the levels of the papillary muscles. In general, the best views were obtained with the transducer lightly applied to the mid-upper left anterior chest wall. The transducer was then gently moved cephalad or caudad and angulated until desirable images were obtained. After it had been ensured that the imaging was on the axis, two-dimensional targeted M-mode tracings were recorded at a paper speed of 50 mm/s.

**Histological analysis.** For histological analysis, LV tissue was fixed in 10% formaldehyde, cut into three transverse sections (apex, middle ring, and base), and then stained with Masson trichrome. Myocyte cross-sectional areas was measured on average of 150 cells from each heart and collagen volume fractions were determined at 5–7 fields for each heart by Image J software (NIH)[33].

**Immunoblot analysis.** Immunoblot analyses were performed using samples from heart or other tissues[34,35]. Samples (i.e., 10–50 μg of total protein from heart or other tissues) were separated by SDS-PAGE and transferred to a polyvinylidene fluoride membrane (Bio-Rad, Herecules, CA). The membrane was blocked for 1 h at room temperature in Tris-buffered saline containing 0.1% Tween 20 (TBS-T) buffer with 3% bovine serum albumin or milk, and then incubated with the primary antibodies (mitoNEET, see "Generation of mitoNEET antibody"; MITOL, ab77585, Abcam, Cambridge, MA; Tom22, ab179826; Tom40, sc-11414, Santa Cruz Biotechnology, Dallas, TX; Tom70, sc-390545; Miner1, #60758, CST, Beverly, MA; 4-hydroxynonenal, ab46545; FtMt, ab111888; MFRN2, ab80467; FXN, ab175402; ABCB7, ab151992; ABCB8, ab133884; TfR, ab84036; DMT1, ab123085; Fpn, ab85370; IRP1, ab126595; IRP2, ab80339; 5′aminolevulinate synthase 1 [ALAS1], ab84962; ferrochelatase [FECH], ab55965; GAPDH, #3683, CST, Beverly, MA) at a dilution of 1:1000 overnight at 4 °C. After three washes with TBS-T, the membrane was incubated with a horseradish peroxide-conjugated secondary antibody at a dilution of 1:5000 for 1 h at room temperature. After washing, the membrane was developed with ECL or ECL Prime Reagent (GE Healthcare Life Sciences, Piscataway, NJ) and then processed for detection with ChemiDoc XRS+ (Bio-Rad). Densities of the bands were quantified using Image J (NIH) software. The expression levels of proteins are shown as values corrected with total protein (CBB staining) or glyceraldehyde phosphate dehydrogenase (GAPDH) expression level.

**Quantitative reverse transcription PCR.** Total RNA was extracted from heart tissues with QuickGene-810 (FujiFilm, Tokyo, Japan) according to the manufacturer's instructions. Total RNA concentrations and purity were assessed by

measuring the optical density (230, 260, and 280 nm) with a Nanodrop 1000 spectrophotometer (Thermo Fisher Scientific, Waltham, MA). cDNA was synthesized with a high-capacity cDNA reverse transcription kit (Applied Biosystems, Carlsbad, CA). Reverse transcription was performed for 10 min at 25 °C, for 120 min at 37 °C, and for 5 s at 85 °C. The solution was then cooled at 4 °C. TaqMan quantitative real-time PCR was performed with a 7300 real-time PCR system (Applied Biosystems) to amplify *Cisd1* (Mm01172641_g1) cDNA from the heart samples. After 2 min at 50 °C and 10 min at 95 °C, PCR amplification was performed for 40 cycles of 15 s at 95 °C and 1 min at 60 °C. GAPDH was used as an internal control. Data were analyzed using the comparative $2^{-\Delta\Delta CT}$ method.

**Preparation of isolated mitochondria.** Heart tissues were quickly harvested, and mitochondria were isolated as described previously[2]. Briefly, heart tissues were minced on ice and incubated with mitochondrial isolation buffer containing 0.1 mg/mL proteinase (Sigma-Aldrich, St. Louis, MO) for 2 min. Heart tissues were gently homogenized with six strokes using a motor-driven Teflon pestle in a glass chamber. The homogenate was centrifuged at $750 \times g$ for 10 min. The supernatant was centrifuged at $10,000 \times g$ for 10 min, and the pellet was washed and centrifuged at $7000 \times g$ for 3 min. The final pellet was suspended in suspension buffer containing 225 mmol/L mannitol, 75 mmol/L sucrose, 10 mmol/L Tris, and 0.1 mmol/L EDTA; pH 7.4. Finally, the mitochondrial protein concentration was measured by the bicinchoninic acid assay.

**Oxidative phosphorylation capacity measurements.** Mitochondrial respiratory capacity was measured in isolated mitochondria from heart tissues at 37 °C with a high-resolution respirometer (Oxygraph-2k, Oroboros Instruments, Innsbruck, Austria) as described previously[34,36,37]. After the addition of isolated mitochondria (approximately 100–200 μg) to the chamber in the respirometer filled with 2 mL of MiR05 medium with 5 U/mL superoxide dismutase, 25 μmol/L Amplex® UltraRed, and 1 U/mL horseradish peroxidase, substrates, adenosine diphosphate (ADP), and inhibitors were added in the following order: [1] glutamate (10 mmol/L) + malate (2 mmol/L) (complex I-linked substrates), [2] ADP (10 mmol/L), [3] succinate (10 mmol/L) (a complex II-linked substrate), [4] oligomycin (2.5 μmol/L), [5] carbonyl cyanide-p-trifluoromethoxyphenylhydrazone (FCCP) (1 μmol/L), [6] rotenone (0.5 μmol/L), and [7] antimycin A, (2.5 μmol/L). $O_2$ consumption rates, i.e., respiratory rates, are expressed as the $O_2$ flux normalized to the mitochondrial protein concentration (μg/μL). Datlab software (Oroboros Instruments) were used for the data acquisition and data analysis. State 2 respiration was assessed by respiration in the presence of exogenous substrates alone. State 3 respiration was assessed by respiration with glutamate, malate, and succinate (complex I- and II-linked substrates) and maximal respiratory capacity was assessed by respiration with FCCP, which is an uncoupler. Reserve capacity was calculated as the difference between the maximal respiratory capacity and state 3 respiration. State 4 respiration was assessed by respiration with oligomycin, which is an inhibitor of ATP synthase.

**Hydrogen peroxide release.** $H_2O_2$ release from isolated mitochondria was measured at 37 °C by spectrofluorometry (O2k-Fluorescence LED2-Module, Oroboros Instruments) as described previously[36]. $H_2O_2$ reacts with Amplex® UltraRed reagent (Thermo Fisher Scientific) at an equal stoichiometry, catalyzed by horseradish peroxidase, which yields the fluorescent compound resorufin (excitation: 560 nm; emission: 590 nm). Resorufin was monitored throughout the experiment. Before the experiment, five different concentrations of $H_2O_2$ were added to establish a standard curve in advance. $H_2O_2$ release rates from isolated mitochondria are expressed as nanomoles per minute per milligram of mitochondrial protein.

**Measurement of mitochondrial iron content.** Isolated mitochondria collected using the Mitochondrial Isolation Kit for Tissue (Pierce, Rockford, IL) were diluted with EDTA-free buffer after sonication. Mitochondrial non-heme iron contents were measured with a commercial Iron Assay Kit (BioAssay Systems, Hayward, CA), which directly detects total iron in the sample, according to the manufacturer's protocol[16]. Isolated mitochondria were sonicated and diluted with EDTA-free buffer, and then iron concentrations were normalized to the mitochondria concentration of each sample.

**Transmission electron microscopy.** The heart muscle sample was fixed in 3% glutaraldehyde with 0.1 mmol/L phosphate buffer and postfixed in 1% osmium tetroxide with 0.1 mmol/L phosphate buffer and then serially dehydrated in ethanol and embedded in epoxy resin. Sections were cut on an ultramicrotome (LKB, Brommer, Sweden) and consecutive ultrathin sections were mounted on copper grids. The ultrathin sections were stained and observed with an electron microscope (H-7100; Hitachi, Tokyo, Japan; JEM1400, JEOL, Tokyo, Japan). The number of mitochondria per field of μm² and the mitochondrial cross-sectional area were quantified. For the determination of cristae density, the outer membrane and the cristae membrane were manually traced, and the sum of the cristae area were divided by the outer area of the mitochondria, as described previously[38]. All measurements were performed by ImageJ 1.52 software.

**Statistics and reproducibility**. The experiments including echocardiography, PCR, oxidative phosphorylation capacity measurements, $H_2O_2$ release, iron content were conducted by researchers (T.F., S. Takada, N.K., S.M., M.T., J.M., W.M., N.E., H.H., J.N.K., T.I., and S. Tanaka) who did not know the group, and finally the data were integrated and analyzed by other researchers (S.K. and A.F.). Quantifications of bands of immunoblotting, histological analysis, and transmission electron microscopic analysis were performed by S.K. and A.F who did not know the group. No statistical methods were used to predetermine sample size. Data are expressed as the mean ± standard error of the mean. The Student unpaired t-test was performed to compare means between two independent groups. ANOVA was performed followed by the Tukey test for multiple-group comparisons of means. Survival analysis was performed by the Kaplan–Meier method, and between-group differences in survival were tested by the log-rank test. P-values less than 0.05 were considered to indicate a statistically significant difference between two groups. All experiments were conducted in at least three independent replicates. All analyses were performed with GraphPad Prism 7 software (GraphPad, La Jolla, CA).

**Reporting summary**. Further information on research design is available in the Nature Research Reporting Summary linked to this article.

## Data availability

Data supporting the findings of this work are available within the paper and its Supplementary files or available from the corresponding author upon reasonable request. Raw data underlying plots in figures are available in Supplementary Data.

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

## Acknowledgements

The authors thank Mami Sato of the Department of Cancer Pathology, Hokkaido University Graduate School of Medicine, for the transmission electron microscopy analysis, and Yuki Kimura, Noriko Ikeda, and Miwako Yamane for their technical assistance. This work was supported in part by Grants-In-Aid for Scientific Research (JP17K15979 [to T.F.], JP17K10137 [to A.F.], JP17H04758 [to S.T.], 18K08022 [to T.Y.], 18H03187 [to S.K.], 26350879 [to S.K.], and 15H04815 [to H.T.]) from the Japan Society for the Promotion of Science.

## Author contributions

T.F., S. Takada, and S.K. designed and conceptualized the study. T.F., S. Takada, N.K., S. Maekawa, M.T., J.M., W.M., N.E., H.H., Y.F., J.N.K., T.I., and S. Tanaka performed the experiments and acquired the data. S.K. and A.F. analyzed the data. T.F., S.K., and H.S. wrote the paper. T.Y., S. Matsushima, H.T., and H.S. edited and revised the manuscript. All authors have read, commented on, and approved the final paper.

## Competing interests

The authors declare no competing interests.
