## [Peer Review File · Communications Biology]

Reviewers' comments:

Reviewer #1 (Remarks to the Author):

Manuscript Summary:

This manuscript report the MitoNEET, a mitochondrial outer member protein, downregulated in the aged heart. The MitoNEET KO mice showed short lifespan and cardiac dysfunction after 12-month old.

Major Concerns:

1. GAPDH is used to normalize the Western blots throughout this study, even though it has been demonstrated elsewhere that GAPDH levels change with age, making it inappropriate to use as a loading control in this context. This makes much of the data difficult to interpret and means that the central observation of MitoNEET decreasing with age and MITOL remaining stable may be incorrect. We recommend instead using the total protein content of each lane on the blot as the loading control or finding several proteins for normalization that have been demonstrated not to change with age.
2. Using only one other mitochondrial outer membrane protein (MITOL; Figure 1) is not sufficient to support the claim that MitoNEET is specifically downregulated with age. It is equally likely that outer membrane proteins are generally downregulated and MITOL levels are specifically preserved. Analysis of additional mitochondrial outer membrane proteins is necessary to make this claim.
3. The heart with 50% decrease of MitoNEET does not show any dysfunction in 12 and 16-month old (Figure 2). Thus, it is hard to draw conclusion that MitoNEET contributes to the natural cardiac dysfunction in aging. Even the homozygous cardiac MitoNEET KO showed some phenotype since 12-month old, this could be caused by the other factors rather than directly from the KO. The causal role of MitoNEET effect on the cardiac natural aging is too weak. The MitoNEET heterozygous with similar expression level at old mouse, could be more helpful to address its aging effect.
4. The mitochondrial irregular morphology in old heart (Figure 4) is unclear. Due to the mitochondria are fixed in the myofilaments, the mito morphology change is very tiny. Some quantification on mitochondrial morphology will be better.
5. The state 3 respiration increases in the WT mitochondria at 16-month old. This contributes a lot to the decrease of the reserve capacity. How about other states respiration and proton leak related respiration? Please show all other states respiration.
6. Even at 3 month old, the MitoNEET KO increased the ROS production. MitoNEET locates at mitochondrial outer member. Whether the mitochondrial electron transport chain complexes function changed in the MitoNEET KO heart?

Minor Concerns:

1. From line 107 to 129, the citations of figure 2 panels are incorrect.
2. Some descriptions are weird. Line 155-156. Line 165-167.

Reviewer #2 (Remarks to the Author):

The manuscript describes the possible role of MitoNEET (*cisd1*) as a role player in heart failure. The paper is well written and timely and recommended to publish with minor recommendations.

1. Abstract: add background strain of KO mice
2. Figure 1. GAPDH may not be best protein marker to standardize since MitoNEET is also mitochondria and there might be changes. suggest to validate with actin or tubulin or discuss why these might not be useful. common practice to use marker opposite of organelle studies.
3. A summarizing figure of possible mechanism would be helpful.
4. The role of iron in mitochondria, and then from the results suggesting its higher in mitochondria, would account for fenton reactions; please discuss with literature some accounts for

effects of iron increase in mitochondria and its role in HF.

5. in the discussion, please speculate on possible therapeutics links with pioglitazone and HF, and if the loss of mitoNEET may then decrease its effectiveness over time.

Reviewer #3 (Remarks to the Author):

MitoNEET is a mitochondrial outer membrane protein that hosts a [2Fe-2S] cluster. Specific function of mitoNEET, however, remains elusive. In this manuscript, the authors reported that during aging, the amount of mitoNEET in heart mitochondria was gradually decreased, which might contribute to heart failure. The authors then constructed a mouse model in which mitoNEET was deleted specifically in heart. The most significant result is shown in Figure 2h, where the mitoNEET-knock-out mice started to die after 15 months and none of them survived after 22 months. In control, wild-type mice were all alive after 22 months. Overall, the results described in the manuscript provided new aspects for further exploring the specific functions of mitoNEET in mitochondria. Nevertheless, there are several concerns that the authors might consider:

1) The authors should at least attempt to speculate the possible mechanisms for the decreased expression of mitoNEET in heart mitochondria in aging mice. For example, mitoNEET has been shown to interact with Parkin, an E3 ubiquitin ligase. Would Parkin be involved in degradation of mitoNEET in heart mitochondria during aging?

2) In addition to mitoNEET, there is a mitoNEET homolog, Miner1, which is also associated with mitochondrial outer membrane (and ER membrane). Both mitoNEET and Miner1 have a high structure similarity. Is there any change of Miner1 in heart mitochondria during aging mice? Could knock-out of mitoNEET in heart mitochondria affect expression of Miner1?

3) Figure 4d shows that knock-out of mitoNEET in heart mitochondria increases the production of H₂O₂ even at age of 3 months. Was there any accumulation of oxidative damage (e.g. membrane lipids or membrane proteins) due to the mitoNEET knock-out in heart mitochondria during aging?

4) Description of Figure 2 in the text was confusing. It seemed that there was an extra panel missing in Figure 2.

Reviewer #4 (Remarks to the Author):

The work described in this manuscript connects the expression level of cardiac mitoNEET to age-related cardiac dysfunction and heart failure. The authors show that mitoNEET protein levels decrease in the hearts of aged mice and describe the generation of a cardiac-specific mitoNEET knockout mouse. Deletion of mitoNEET resulted in increased cardiac impairment with aging and earlier death by heart failure. To further probe the cause of impairment in heart function the authors used microscopy and fluorescence-based assays to evaluate mitochondrial structure and function in the aged mice. They found that in the knockout mice mitochondrial morphology was altered and respiratory function was reduced with aging. Additionally, mitochondrial iron content as well as peroxide release were increased in both the 3 month old and 12 month old mice. These results provide strong evidence for the role of decreased mitoNEET expression in age-related cardiac dysfunction. This study would be of interest to researchers studying mechanisms of aging as well as cardiac dysfunction. Overall, the manuscript is well written and the data support the authors' conclusions.

Comments:

1. Knockout of cardiac mitoNEET causes increased mitochondrial iron content with increased

mitochondrial ferritin. There are several biochemical studies (see publications from Jennings, Nechushtai and Bouton) on the redox dependent ability of mitoNEET to transfer iron/iron-sulfur clusters to other proteins. The authors should include some background and references on this topic to connect loss of mitoNEET to mitochondrial iron accumulation.

2. It would be helpful to the reader to briefly describe the connection of mitochondrial iron accumulation with increased reactive oxygen species generation.

Responses to the Comments from Reviewer #1

We greatly thank the reviewer for the helpful comments concerning improvements of our manuscript. The followings are our responses to each of your comments.

Major Concerns:

1. GAPDH is used to normalize the Western blots throughout this study, even though it has been demonstrated elsewhere that GAPDH levels change with age, making it inappropriate to use as a loading control in this context. This makes much of the data difficult to interpret and means that the central observation of MitoNEET decreasing with age and MITOL remaining stable may be incorrect. We recommend instead using the total protein content of each lane on the blot as the loading control or finding several proteins for normalization that have been demonstrated not to change with age.

Author's response:

We appreciate this important comment. As the Reviewer suggested, we evaluated the total protein by Coomassie Brilliant Blue (CBB) staining and used it as the loading control in Figure 1 and Supplementary Figure 1. We revised the description of loading control in the Results section (Page 6, lines 89-90) and added the method of CBB staining in the Methods section (Page 18, lines 364-366). Please also see our response to the comment 2 from the Reviewer #2.

(Results section; Page 6, lines 89-90)

We found that mitoNEET levels are significantly decreased in the hearts of 12-month-old C57BL/6J mice compared with those of 3-month-old mice, in which relative protein levels were normalized to total protein evaluated by Coomassie Brilliant Blue (CBB) staining (Figure 1).

(Methods section: Page 18, lines 364-366)

Densities of the bands were quantified using Image J (NIH) software. The expression levels of proteins are shown as values corrected with total protein (CBB staining) or GAPDH expression level.

Figure 1.

Supplementary Figure 1.

2. Using only one other mitochondrial outer membrane protein (MITOL; Figure 1) is not sufficient to support the claim that MitoNEET is specifically downregulated with age. It is equally likely that outer membrane proteins are generally downregulated and MITOL levels are specifically preserved. Analysis of additional mitochondrial outer membrane proteins is necessary to make this claim.

Author's response:

We appreciate this helpful comment. We completely agree with the Reviewer. We evaluated the expression levels of other outer mitochondrial membrane protein such as Tom22, Tom40, and Tom70. These proteins did not show an age-dependent decrease (Figure 1). We added these results in the Results section (Page 6, line 90) and added information of these antibodies in the Methods section (Page 18, lines 353-354). Please also see our response to comment 2 from the Reviewer #3.

(Results section; Page 6, line 90)

MITOL, Tom22, Tom40, and Tom70, which are outer mitochondrial membrane protein like mitoNEET, did not show such a decrease (**Figure 1**).

(Methods section; Page 18, lines 353-354)

mitoNEET, see *Generation of mitoNEET Antibody*; MITOL, ab77585, Abcam, Cambridge, MA; Tom22, ab179826; Tom40, sc-11414, Santa Cruz Biotechnology, Dallas, TX; Tom70, sc-390545; Miner1, #60758, CST, Beverly, MA; 4-hydroxynonenal, ab46545; FtMt, ab111888; MFRN2, ab80467;

3. The heart with 50% decrease of MitoNEET does not show any dysfunction in 12 and 16-month old (Figure 2). Thus, it is hard to draw conclusion that MitoNEET contributes to the natural cardiac dysfunction in aging. Even the homozygous cardiac MitoNEET KO showed some phenotype since 12-month old, this could be caused by the other factors rather than directly from the KO. The causal role of MitoNEET effect on the cardiac natural aging is too weak. The MitoNEET heterozygous with similar expression level at old mouse, could be more helpful to address it is aging effect.

Author's response:

We appreciate this crucial comments. As the Reviewer suggested, 12- or 16-month old WT mice on C57BL/6 background, which probably have 50% decrease of mitoNEET protein in the heart, did not show cardiac and mitochondrial dysfunction (Figures 2 and 3). The mitoNEET homozygous KO mice in which the mitoNEET was completely deficient in the heart began to show impaired mitochondrial function and heart function after 12 months of age (Figures 2 and 3). These results suggest that mitoNEET does not directly regulate mitochondrial function or cardiac function, but is involved in regulation of mitochondrial membrane integrity and iron within mitochondria, and as the result, gradually affects mitochondrial function and cardiac function. Therefore, it is consistent with our results that 12- or 16-month old WT mice, which have a heart with 50% decrease of mitoNEET, do not exhibit cardiac dysfunction. Unfortunately, we could not decide the precise role of mitoNEET and the causal relationship between mitoNEET and cardiac dysfunction in normal aging, but our results suggest that the gradual decrease in mitoNEET in the heart could be associated with cardiac dysfunction in the later phase of normal aging. Therefore, we revised the title to “Naturally occurring mitoNEET downregulation in aged hearts is closely associated with age-related heart failure”.

Furthermore, it is important to investigate the dose-dependent role of mitoNEET on aging process using the mitoNEET heterozygous KO mice. However, studies using the mitoNEET heterozygous would require an observation period of nearly 2 years, and we will try to perform the study as future plan.

(Title; Page 1, lines 1-2)

Naturally occurring mitoNEET downregulation in aged hearts is closely associated with age-related heart failure

4. The mitochondrial irregular morphology in old heart (Figure 4) is unclear. Due to the mitochondria are fixed in the myofilaments, the mito morphology change is very tiny. Some quantification on mitochondrial morphology will be better.

Author's response:

We appreciate this helpful comments. We quantified cristae density as an established index of mitochondrial irregular morphology. Cristae density was significantly decreased in KO mice compared to controls at 3 and 12 months of age. We added these

data in the Results section (Page 9, lines 170-171) and in Figure 4, and we also revised the method of transmission electron microscopy in the Methods section (Page 22, lines 443-447) with additional reference (Reference No. 35).

(Methods section; Page 22, lines 443–447)

The ultrathin sections were stained and observed with an electron microscope (H-7100, Hitachi, Tokyo, Japan; JEM1400, JEOL, Tokyo, Japan). The number of mitochondria per field of μm^2 and the mitochondrial cross-sectional area were quantified. For the determination of cristae density, the outer membrane and the cristae membrane were manually traced, and the sum of the cristae area were divided by the outer area of the mitochondria, described previously³⁵. All measurements were performed by ImageJ 1.52 software.

(Results section; Page 9, lines 170–171)

Transmission electron microscopic analysis demonstrated no notable differences between the *mitoNEET* KO mice and controls with regard to the number of mitochondria and their cross-sectional areas, when analyzed at 3, 12, and 16 months of age (**Figure 4**). In contrast, cristae density was significantly decreased in KO mice compared to controls, when measured at 3 and 12 months of age (**Figure 4**). Morphological integrity, as well as integrity of the outer membrane of mitochondria, is crucial to avoid the unnecessary leakage of electrons from the respiratory chain, that otherwise leads to the generation of ROS^{3, 19}.

Figure 4.

5. The state 3 respiration increases in the WT mitochondria at 16-month old. This contributes a lot to the decrease of the reserve capacity. How about other states respiration and proton leak related respiration? Please show all other states respiration.

Author's response:

We appreciate this important comments. As the Reviewer pointed out, the state 3 respiration seemed to be increased in WT at 16 months of age compared with WT at younger ages, although we did not perform formal statistical analysis. This may indicate the decreased amount of mitochondria or the poor efficiency of mitochondrial respiration during normal aging. According to the Reviewer's suggestion, we showed the data of all states respiration in Figure 3. There were no significant differences in state 2 between controls and mitoNEET KO mice at all ages. State 4 was slightly decreased in mitoNEET KO mice compared with controls, when they were older than 12 months. We added these data in the Results section (Page 9, lines 161-164) and in Figure 3, and revised oxidative phosphorylation capacity measurements in the Methods section (Page 20, lines 408-409, and lines 413-414).

(Results section; Page 9, lines 161-164)

Therefore, the loss of mitoNEET from birth caused detrimental effects on mitochondrial respiration only when mice became old, as we observed in the 12-month-old KO mice. On the other hand, there were no significant differences in state 2 respiration between controls and KO mice at all ages (Figure 3). State 4 respiration was slightly decreased in KO mice compared with controls, when they were older than 12 months (Figure 3).

(Methods section; Page 9, lines 408-409 and lines 413-414)

State 2 respiration was assessed by respiration in the presence of exogenous substrates alone. State 3 respiration was assessed by respiration with glutamate, malate, and succinate (complex I- and II-linked substrates) and maximal respiratory capacity was assessed by respiration with FCCP, which is an uncoupler. Reserve capacity was calculated as the difference between the maximal respiratory capacity and state 3 respiration. State 4 respiration was assessed by respiration with oligomycin, which is an inhibitor of ATP synthase.

Figure 3.

6. Even at 3 month old, the MitoNEET KO increased the ROS production.

MitoNEET locates at mitochondrial outer member. Whether the mitochondrial electron transport chain complexes function changed in the mitoNEET KO heart?

Author's response:

We appreciate this helpful comments. There were no significant differences in all states respiration in between mitoNEET KO mice and controls at 3 months of age (**Figure 3a**). Therefore, there was no significant increase in proton leak, and complex I and II-linked oxidative phosphorylation capacity and maximal electron transfer chain capacity were preserved in mitoNEET KO mice at 3 months of age, which suggest that all complex

activities including complex I-IV remained normal.

Under the condition of mitochondrial iron overload with the deletion of mitoNEET, an increase in H₂O₂ release was detected (**Figure 5a, b**). Iron plays a crucial role in the redox reaction *in vivo*, and its overload can cause free radical production through many pathways via a reduction of oxygen. Moreover, reactive oxygen species (ROS) are generated by means of the Fenton reaction in the presence of endogenous iron. Therefore, a mitochondrial iron overload can easily enhance superoxide production via oxidative phosphorylation, even if the overall mitochondrial function is preserved (**Figure 3a**). This was consistent with the overall mitochondrial function and cardiac function in the 3-month-old mice (**Figure 2a-d**). These results suggest that disruption of mitoNEET primarily causes a mitochondrial iron overload and enhances ROS production, which secondarily leads to mitochondrial dysfunction. We added the description about this crucial point in the Discussion section (Page 13, lines 236-246). Please also see our responses to the comment 4 from the Reviewer #2 and the comment 2 from the Reviewer #4.

(Discussion section; Page 13, lines 236-246)

Under the condition of mitochondrial iron overload with the deletion of mitoNEET, an increase in H₂O₂ release was detected. Iron plays a crucial role in the redox reaction *in vivo*, and its overload can cause free radical production through many pathways via a reduction of oxygen. Moreover, ROS are generated by means of the Fenton reaction in the presence of endogenous iron. Therefore, a mitochondrial iron overload can easily enhance superoxide production via oxidative phosphorylation, even if the overall mitochondrial function is preserved. This was consistent with the overall mitochondrial function and cardiac function in the 3-month-old mice. These results suggest that disruption of mitoNEET primarily causes a mitochondrial iron overload and enhances ROS production, which secondarily leads to mitochondrial dysfunction. Enhanced ROS production leads to cardiac dysfunction and the development of HF.

Minor Concerns:

1. From line 107 to 129, the citations of figure 2 panels are incorrect

Author's response:

In response, we revised them.

2. Some description are weird. Line 155-156. Line 165-167

Author's response:

In response, we revised them.

Again, we appreciate the opportunity to revise our manuscript for consideration of publication in *Communications Biology*. We hope our revisions adequately address the concerns raised by the reviewers and hope that the revised manuscript can now again be considered for publication.

Responses to the Comments from Reviewer #2

We greatly thank the reviewer for the helpful comments concerning improvements of our manuscript. The followings are our responses to each of your comments.

Comments:

1. Abstract: add background strain of KO mice

Author's response:

In response, we added information of background strain of KO mice in the Abstract section (Page 3, lines 37-38).

(Abstract section; Page 3, lines 37-38)

Mice with a constitutive cardiac-specific deletion of *CISDI* on the C57BL/6J background showed cardiac dysfunction only after 12 months of age and developed HF after 16 months; whereas irregular morphology and higher levels of reactive oxygen species in their cardiac mitochondria were observed at earlier time points.

2. Figure 1. GAPDH may not be best protein marker to standardize since mitoNEET is also mitochondria and there might be changes. suggest to validate with actin or tubulin or discuss why these might not be useful. common practice to use marker opposite of organelle studies.

Author's response:

We appreciate this important comment. As the Reviewer suggested, we evaluated the total protein by CBB staining and used it as the loading control in Figure 1 and Supplementary Figure 1. We revised the description of loading control in the Results section (Page 6, lines 89-90) and added the method of CBB staining in the Methods section (Page 18, lines 364-367). Please also see our response to the comment 1 from the Reviewer #1.

(Results section; Page 6, lines 89-90)

We found that mitoNEET levels are significantly decreased in the hearts of 12-month-old C57BL/6J mice compared with those of 3-month-old mice, in which relative protein levels were normalized to total protein evaluated by Coomassie Brilliant

Blue (CBB) staining (**Figure 1**).

(**Methods section: Page 18, lines 364-367**)

Densities of the bands were quantified using Image J (NIH) software. The expression levels of proteins are shown as values corrected with total protein (CBB staining) or GAPDH expression level.

Figure 1.

3. A summarizing figure of possible mechanism would be helpful.

Author's response:

We appreciate this helpful suggestion. We added a summarizing figure of possible mechanism in the present study in Figure 6.

Figure 6.

4. The role of iron in mitochondria, and then from the results suggesting its higher in mitochondria, would account for fenton reactions; please discuss with literature some accounts for effects of iron increase in mitochondria and its role in HF.

Author's response:

We appreciate this helpful comments. We agree with the Reviewer's suggestion. We added important discussion about iron overload in mitochondria in mitoNEET KO mice, and ROS production through Fenton reaction. Our results suggest that disruption of mitoNEET primarily causes a mitochondrial iron overload and enhances ROS production, which secondarily leads to mitochondrial dysfunction. Furthermore, based on previous papers, we discussed that ROS production played an important role in cardiac dysfunction and heart failure. We added crucial issues in the Discussion section (Page 13, lines 236-253) with additional references (Reference No. 25-27). Please also see our responses to the comment 6 from the Reviewer #1 and the comment 2 from the Reviewer #4.

(Discussion section; Page 13, lines 236–253)

Under the condition of mitochondrial iron overload with the deletion of mitoNEET, an increase in H₂O₂ release was detected. Iron plays a crucial role in the redox reaction *in vivo*, and its overload can cause free radical production through many pathways via a reduction of oxygen. Moreover, ROS are generated by means of the Fenton reaction in the presence of endogenous iron. Therefore, a mitochondrial iron overload can easily enhance superoxide production via oxidative phosphorylation, even if the overall mitochondrial function is preserved. This was consistent with the overall mitochondrial function and cardiac function in the 3-month-old mice. These results suggest that disruption of mitoNEET primarily causes a mitochondrial iron overload and enhances ROS production, which secondarily leads to mitochondrial dysfunction. Many investigations have shown that enhanced ROS production leads to cardiac dysfunction and the development of HF. We reported that the exposure of cardiac myocytes to H₂O₂ to leads to their injury²⁵. We also demonstrated that mitochondria-derived ROS production was increased in hearts from HF model mice²⁶, and that treatment with an antioxidant and the overexpression of a mitochondrial antioxidant (such as peroxiredoxin-3) improved cardiac function and HF²⁷. Therefore, a long-term exposure

of ROS over the physiological level could lead to cardiac dysfunction.

5. in the discussion, please speculate on possible therapeutics links with pioglitazone and HF, and if the loss of mitoNEET may then decrease its effectiveness over time.

Author's response:

We appreciate this helpful comment. mitoNEET was originally identified as a target of the insulin-sensitizing drug pioglitazone. It has been known that binding of pioglitazone stabilizes mitoNEET against 2Fe-2S cluster release, which are observed under oxidized condition (reference No. 10, 12). Pioglitazone can inhibit iron transfer from mitoNEET to mitochondria, and iron overload within mitochondria (reference No. 12). Furthermore, pioglitazone treatment for spinal cord injury mice improved state 3 respiration and electron transport system capacity of mitochondria in neuron cells, and these effects of pioglitazone were not observed in mitoNEET KO mice (reference No. 28). Therefore, our data suggest that long-term treatment with pioglitazone may prevent age-associated cardiac dysfunction and heart failure by stabilizing mitoNEET, and age-associated mitoNEET downregulation may weaken its effectiveness. We now added this description on clinical implication in the Discussion section (Page 13, line 254-Page 14, line 264) with additional references (Reference No. 10, 12, and 28).

(Discussion section; Page 13, line 254-Page 14, line 264)

mitoNEET was originally identified as a target of the insulin-sensitizing drug pioglitazone. It has been known that binding of pioglitazone stabilizes mitoNEET against 2Fe-2S cluster release, which are observed under oxidized condition^{10, 12}. Pioglitazone can inhibit iron transfer from mitoNEET to mitochondria, and iron overload within mitochondria¹². Furthermore, pioglitazone treatment for spinal cord injury mice improved state 3 respiration and electron transport system capacity of mitochondria in neuron cells, and these effects of pioglitazone were not observed in mitoNEET KO mice²⁸. Therefore, our data suggest that long-term treatment with pioglitazone may prevent age-associated cardiac dysfunction and heart failure by stabilizing mitoNEET, and age-associated mitoNEET downregulation may weaken its effectiveness.

Again, we appreciate the opportunity to revise our manuscript for consideration of publication in *Communications Biology*. We hope our revisions adequately address the concerns raised by the reviewers and hope that the revised manuscript can now again be considered for publication.

Responses to the Comments from Reviewer #3

We greatly thank the reviewer for the helpful comments concerning improvements of our manuscript. The followings are our responses to each of your comments.

Comments:

1) The authors should at least attempt to speculate the possible mechanisms for the decreased expression of mitoNEET in heart mitochondria in aging mice. For example, mitoNEET has been shown to interact with Parkin, an E3 ubiquitin ligase. Would Parkin be involved in degradation of mitoNEET in heart mitochondria during aging?

Author's response:

We appreciate this helpful comment. We could not clarify the precise mechanism involved in the age-associated mitoNEET downregulation. Recent paper reported significant increases in Parkin, an E3 ubiquitin ligase, and ubiquitinated proteins in the mitochondrial fraction from hearts of C57BL/6 mice at 24 months of age (reference No. 20). mitoNEET has been shown to be a direct substrate of Parkin and to be polyubiquitinated by Parkin (reference No. 21). Therefore, Parkin may be involved in degradation of mitoNEET in aged hearts, however, the functional implications of this translational modification of mitoNEET are unknown. We added these descriptions in the Discussion section (Page 12, lines 214-220).

(Discussion section; Page 12, lines 214-220)

However, the precise mechanism involved in this age-associated mitoNEET downregulation still remains unclear, although we observed that *mitoNEET* mRNA levels were slightly decreased in the hearts of aged mice (our unpublished results). Recent paper reported significant increases in Parkin, an E3 ubiquitin ligase, and ubiquitinated proteins in the mitochondrial fraction from hearts of C57BL/6 mice at 24 months of age²⁰. mitoNEET has been shown to be a direct substrate of Parkin and to be polyubiquitinated by Parkin²¹. Therefore, Parkin may be involved in degradation of mitoNEET in aged hearts, however, the functional implications of this translational modification of mitoNEET are unknown. On the other hand, it is noteworthy that a similar downregulation of mitoNEET is seen in the kidney but not in other tissues and

organs.

2) In addition to mitoNEET, there is a mitoNEET homolog, Miner1, which is also associated with mitochondrial outer membrane (and ER membrane). Both mitoNEET and Miner1 have a high structure similarity. Is there any change of Miner1 in heart mitochondria during aging mice? Could knock-out of mitoNEET in heart mitochondria affect expression of Miner1?

Author's response:

We appreciate this helpful comments. We evaluated Miner1 expression level in the heart during aging. It did not show an age-dependent decrease (Figure 1). We also evaluated Miner1 expression level in the heart from mitoNEET KO mice compared with WT mice. Again, there was no significant difference in Miner1 expression level between these mice (Supplementary Figure 2g). We added these results in the Results section (Page 6, lines 92-93, and Page 7, lines 113-114) and information of the antibody in the Methods section (Page 18, line 354). Please also see our response to comment 2 from the Reviewer #1.

(Results section; Page 6, lines 92-93)

MITOL, Tom22, Tom40, and Tom70, which are outer mitochondrial membrane protein like mitoNEET, did not show such a decrease (**Figure 1**). Miner1, which is a mitoNEET homolog and is localized to the mitochondria-associated membrane, also did not change during aging (Figure 1). A similar age-dependent decrease in mitoNEET levels was also observed in the kidney, but not in the liver, the skeletal muscle, or the brain (**Supplementary Figure 1**).

(Results section; Page 7, lines 113-114)

Immunoblot and immunohistochemical analysis also confirmed absence of the mitoNEET protein in cardiac cells of KO mice (**Supplementary Figure 2d, e**), whereas it was expressed in the brain, heart, liver, kidney, and skeletal muscle of KO mice, as in control mice (**Supplementary Figure 2f**). There was no significant difference in the expression of Miner1 in the heart between KO and control (Supplementary Figure 2g). These cardiac-specific *mitoNEET* KO mice were viable and fertile, and there were no notable differences in appearance and body weight between KO mice and control mice

(Supplementary Table 1).

(Methods section; Page 18, line 354)

MITOL, ab77585, Abcam, Cambridge, MA; Tom22, ab179826; Tom40, sc-11414, Santa Cruz Biotechnology, Dallas, TX; Tom70, sc-390545; Miner1, #60758, CST, Beverly, MA; 4-hydroxynonenal, ab46545; FtMt, ab111888; MFRN2, ab80467;

Figure 1.

Supplementary Figure 2g

3) Figure 4d shows that knock-out of mitoNEET in heart mitochondria increases the production of H₂O₂ even at age of 3 months. Was there any accumulation of oxidative damage (e.g. membrane lipids or membrane proteins: TBARS) due to the mitoNEET knock-out in heart mitochondria during aging?

Author's response:

We appreciate this helpful comment. According to the Reviewer's suggestion, we evaluated 4-hydroxy-2-nonenal (HNE), an aldehydic byproduct of lipid peroxidation, in the heart of controls and mitoNEET KO mice during aging. HNE was increased in the heart from mitoNEET KO mice compared with controls both at 3 months and 12 months of age (Figure 5c, d). Changes in HNE was consistent with the production of H₂O₂. Long-term accumulation of oxidative damage due to excessive production of ROS would be associated with the dysfunction of mitochondrial respiration and LV function. We added these results in the Results section (Page 10, lines 179-185), and added information of the antibody in the Methods section (Page 18, line 354).

(Results section; Page 10, lines 179-185)

Together with the above results, these results collectively suggested that the loss of mitoNEET may primarily affects the morphological integrity of mitochondria, and this becomes evident at much earlier ages in mice than the dysfunction of mitochondrial respiration and LV function. We also evaluated 4-hydroxy-2-nonenal (HNE), an aldehydic byproduct of lipid peroxidation, in the heart of controls and mitoNEET KO mice during aging. HNE was increased in the heart from mitoNEET KO mice compared with controls both at 3 months and 12 months of age (Figure 5c, d). Changes in HNE was consistent with the production of H₂O₂. Long-term accumulation of oxidative damage due to excessive production of ROS would be associated with the dysfunction of mitochondrial respiration and LV function.

(Methods section; Page 18, line 354)

MITOL, ab77585, Abcam, Cambridge, MA; Tom22, ab179826; Tom40, sc-11414, Santa Cruz Biotechnology, Dallas, TX; Tom70, sc-390545; Miner1, #60758, CST, Beverly, MA; 4-hydroxynonenal, ab46545; FtMt, ab111888; MFRN2, ab80467;

Figure 5c, d

4) Description of Figure 2 in the text was confusing. It seemed that there was an extra panel missing in Figure 2.

Author's response:

In response, we revised the description of Figure 2 in the text (Page 7, line 118-Page 8, line 141)

Again, we appreciate the opportunity to revise our manuscript for consideration of publication in Communications Biology. We hope our revisions adequately address the concerns raised by the reviewers and hope that the revised manuscript can now again be considered for publication.

Responses to the Comments from Reviewer #4

We greatly thank the reviewer for the helpful comments concerning improvements of our manuscript. The followings are our responses to each of your comments.

Comments:

1. Knockout of cardiac mitoNEET causes increased mitochondrial iron content with increased mitochondrial ferritin. There are several biochemical studies (see publications from Jennings, Nechushtai and Bouton) on the redox dependent ability of mitoNEET to transfer iron/iron-sulfur clusters to other proteins. The authors should include some background and references on this topic to connect loss of mitoNEET to mitochondrial iron accumulation.

Author's response:

We appreciate this helpful comment. According to the Reviewer's suggestion, we added the description on the role of mitoNEET in mitochondrial iron metabolism in the Introduction section (Page 4, lines 61-65) with additional references (Reference No. 11-15).

(Introduction section; Page 4, lines 61-65)

MitoNEET contains redox-sensitive 2Fe-2S clusters, and under oxidized state, it can transfer its 2Fe-2S clusters to an apo-acceptor protein¹¹⁻¹³. Recently, it has been shown that mitoNEET plays a critical role in cytosolic Fe-S cluster repair of iron-regulatory rotein-1¹⁴. mitoNEET can also regulates free iron level within mitochondria by regulating the channel function of voltage-dependent anion channel 1¹⁵. The systemic inducible knockdown of mitoNEET was found to cause mitochondrial iron overload in adipocytes and liver, and increased iron-mediated mitochondrial lipid peroxidation.

2. It would be helpful to the reader to briefly describe the connection of mitochondrial iron accumulation with increased reactive oxygen species generation.

Author's response:

We appreciate this helpful comments. We added important discussion about iron overload in mitochondria in mitoNEET KO mice, and ROS production through Fenton

reaction. Our results suggest that disruption of mitoNEET primarily causes a mitochondrial iron overload and enhances ROS production, which secondarily leads to mitochondrial dysfunction. Furthermore, based on previous papers, we discussed that ROS production played an important role in cardiac dysfunction and heart failure. We added crucial issues in the Discussion section (Page 13, lines 236-253) with additional references (Reference No. 25-27). Please also see our responses to the comment 6 from the Reviewer #1 and the comment 4 from the Reviewer #2.

(Discussion section; Page 13, lines 236–253)

Under the condition of mitochondrial iron overload with the deletion of mitoNEET, an increase in H₂O₂ release was detected. Iron plays a crucial role in the redox reaction *in vivo*, and its overload can cause free radical production through many pathways via a reduction of oxygen. Moreover, ROS are generated by means of the Fenton reaction in the presence of endogenous iron. Therefore, a mitochondrial iron overload can easily enhance superoxide production via oxidative phosphorylation, even if the overall mitochondrial function is preserved. This was consistent with the overall mitochondrial function and cardiac function in the 3-month-old mice. These results suggest that disruption of mitoNEET primarily causes a mitochondrial iron overload and enhances ROS production, which secondarily leads to mitochondrial dysfunction. Many investigations have shown that enhanced ROS production leads to cardiac dysfunction and the development of HF. We reported that the exposure of cardiac myocytes to H₂O₂ to leads to their injury²⁵. We also demonstrated that mitochondria-derived ROS production was increased in hearts from HF model mice²⁶, and that treatment with an antioxidant and the overexpression of a mitochondrial antioxidant (such as peroxiredoxin-3) improved cardiac function and HF²⁷. Therefore, a long-term exposure of ROS over the physiological level could lead to cardiac dysfunction.

Again, we appreciate the opportunity to revise our manuscript for consideration of publication in Communications Biology. We hope our revisions adequately address the concerns raised by the reviewers and hope that the revised manuscript can now again be considered for publication.

REVIEWERS' COMMENTS:

Reviewer #1 (Remarks to the Author):

The authors addressed my comments in the revision.

Reviewer #2 (Remarks to the Author):

The revised manuscript describes the effect of mitoNEET, a mitochondrial protein on cardiovascular health, specifically in the heart, as it relates to age. To date, no literature has been published on the aging effect of mitoNEET in naturally occurring C57BL/6 mice, although KO studies in heart and brain has indicated an important role in aging diseases. The studies give a new perspective on iron regulation in the heart. These results will lead to novel insight into new treatment modalities of cardiac disease.

Minor comment: Page 15. please add that tissues were dissected for analysis, as brain was also used, and not just heart and kidney. Also, please mention that whole brain was used. This gives a general average insight into the brain, but needs to be mentioned since brain regions have different phenotypical effects due to mitochondrial dysfunction for instance the striatum is sensitive to iron dysregulation.

Reviewer #3 (Remarks to the Author):

The authors carefully addressed the concerns.

A minor point. As the authors indicated, the physiological function of mitoNEET is still not fully understood. If ROS production due to depletion of mitoNEET could be the main course for the observed dysfunction of mitochondria (page 13), it is highly possible that mitoNEET may act as an electron transfer protein (page 5, line 68) participating in ROS metabolism in mitochondria. In this context, a number of related papers have been published focusing on the electron transfer activity of mitoNEET (e.g. Landry et al,2017, FRBM; Wang, et al, 2017, JBC; Tasnim, et al, 2020,FRBM).

Reviewer #4 (Remarks to the Author):

The additional data and changes to the text adequately address the reviewers' concerns and improve the quality and clarity of the manuscript. The addition of the CBB total protein staining control as well as the analysis of additional mitochondrial protein levels addresses an important concern regarding the use of appropriate protein level controls. This strengthens the authors' conclusions regarding the decrease in mitoNEET levels in the hearts of aged mice.

Responses to the Comments from Reviewer #2

We greatly thank the reviewer for the helpful comments concerning improvements of our manuscript. The followings are our responses to each of your comments.

Minor Comment:

Page 15. Please add that tissues were dissected for analysis, as brain was also used, and not just heart and kidney. Also, please mention that whole brain was used. This gives a general average insight into brain, but needs to be mentioned since brain regions have different phenotypical effects due to mitochondrial dysfunction for instance the striatum is sensitive to iron dysregulation.

Author's response:

In response, we added this description (Page 15, line 294).

(Methods section; Page 15, line 294)

Male C57BL/6J mice (CLEA Japan, Tokyo) were euthanized under deep anesthesia with an overdose of pentobarbital (100 mg/kg i.p.) at the age of 3, 6, and 12 months. The heart, kidney, liver, gastrocnemius muscle, and whole brain were excised, and used for immunoblot analysis.

Again, we appreciate the opportunity to revise our manuscript for consideration of publication in *Communications Biology*. We hope our revisions adequately address the concerns raised by the reviewers and hope that the revised manuscript can now again be considered for publication.

Responses to the Comments from Reviewer #3

We greatly thank the reviewer for the helpful comments concerning improvements of our manuscript. The followings are our responses to each of your comments.

Minor Point:

As the authors indicated, the physiological function of mitoNEET is still not fully understood. If ROS production due to depletion of mitoNEET could be the main course for the observed dysfunction of mitochondria (page 13), it is highly possible that mitoNEET may act as an electron transfer protein (Page 5, line 68) participating in ROS metabolism in mitochondria. In this context, a number of related papers have been published focusing on the electron transfer activity of mitoNEET (e.g. Landry et al, 2017, FRBM; Wang et al, 2017, JBC; Tasnim et al, 2020, FRBM).

Author's response:

We appreciate this helpful suggestion. We completely agree with the Reviewer's suggestion and added discussion on this issue with additional references (reference No 25-27) in the discussion section (Page 13, lines 243-248).

(Discussion section; Page 13, lines 243-248)

This was consistent with the overall mitochondrial function and cardiac function in the 3-month-old mice. Furthermore, it has been known that mitoNEET functions as electron-transfer protein^{9, 25-27}. The reduced flavin mononucleotide interacts with mitoNEET via specific binding site and transfers its electrons to the 2Fe-2S clusters of mitoNEET, and the reduced 2Fe-2S clusters in mitoNEET transfer the electrons to oxygen or ubiquinone. Therefore, mitoNEET may participate in ROS metabolism in mitochondria. These suggest that disruption of mitoNEET primarily causes a mitochondrial iron overload and enhances ROS production, which secondarily leads to mitochondrial dysfunction (**Figure 6**).

Again, we appreciate the opportunity to revise our manuscript for consideration of publication in Communications Biology. We hope our revisions adequately address the concerns raised by the reviewers and hope that the revised manuscript can now again be

considered for publication.